# Treatment Experience and Predictive Factors Associated with Response in Platinum-Resistant Recurrent Ovarian Cancer: A Retrospective Single-Institution Study

**DOI:** 10.3390/jcm10163596

**Published:** 2021-08-15

**Authors:** Radu Dragomir, Ioan Sas, Sorin Săftescu, Dorel Popovici, Roxana Margan, Adelina Silvana Dragomir, Horia Stanca, Valeria Mocanu, Cristina Pac, Șerban Negru

**Affiliations:** 1Department of Obstetrics and Gynecology, “Victor Babeș” University of Medicine and Pharmacy, 300041 Timisoara, Romania; dragomirradu91@gmail.com; 2Department of Oncology, “Victor Babeș” University of Medicine and Pharmacy, 300041 Timisoara, Romania; sorin251@yahoo.com (S.S.); dorelpopovici@gmail.com (D.P.); snegru@yahoo.com (Ș.N.); 3Doctoral School, “Victor Babeș” University of Medicine and Pharmacy, 300041 Timisoara, Romania; marganroxana@gmail.com; 4Department of Oncology, “Carol Davila” University of Medicine and Pharmacy, 020021 Bucharest, Romania; adelina.silvana.gheorghe@gmail.com; 5Department of Ophthalmology, “Carol Davila” University of Medicine and Pharmacy, 020021 Bucharest, Romania; hstanca@yahoo.com; 6Department of Ophthalmology, “Victor Babeș” University of Medicine and Pharmacy, 300041 Timisoara, Romania; valeria_mocanu@yahoo.com (V.M.); kittysmileface@yahoo.com (C.P.)

**Keywords:** ovarian cancer, platinum resistance, predictive factors

## Abstract

Ovarian cancer (OC) represents the most common and lethal gynecologic malignancy, due to its increased incidence and mortality rate. It is usually diagnosed in advanced stages and, even though surgery and platinum-based treatments are initially efficient, recurrences emerge in over 70% of cases. Although there are multiple options of chemotherapy drugs from which to choose, little is known regarding the best strategy for prolonged survival. Thus, this study aimed to assess the effect that most frequently used chemotherapeutic regimens have upon time-to-treatment-failure (TTF) from the first line and beyond, considering clinical and biological factors which influence the treatment outcome of platinum-resistant recurrent OC. We retrospectively analyzed data from 78 patients diagnosed with platinum-resistant OC, who underwent chemotherapy-based treatment with or without anti-angiogenic therapy at OncoHelp Oncology Center, Romania (January 2016–February 2021). Our study identified positive predictive factors for TTF related to histology (serous carcinoma subtype), anthropometry (age over 60 for patients treated with topotecan with or without bevacizumab), renal function (creatinine levels between 0.65 and 1 mg/dL for patients treated with regimens containing bevacizumab and pegylated liposomal doxorubicin) and treatment choice (bevacizumab in combination with pegylated liposomal doxorubicin or topotecan used from the first line and beyond).

## 1. Introduction

Ovarian cancer (OC) remains one of the most common and deadly gynecological malignancies, with epidemiological discrepancies around the world. According to Globocan, in 2020, 313,959 new cases of ovarian cancer were reported worldwide, with 207,252 deaths, thus showing an estimated mortality–incidence rate of 2/3 [1]. OC presents three main histological types, with epithelial being the most frequent. This type, in turn, divides into four subtypes: serous (with its two forms: high-grade and low-grade), endometrioid, mucinous, and clear-cell [2]. Not only are most ovarian cancer patients diagnosed at an advanced stage (Féderation Internationale de Gynécologie et d’Obstétrique (FIGO) III–IV), which leads to such a poor outcome, but over 70% present with a recurrence of the disease, despite response at initial treatment with surgery and platinum-based chemotherapy [3,4]. Novel targeted therapies, like anti-angiogenic drugs and Poly (ADP-ribose) polymerase (PARP) inhibitors were approved in the past decade, showing promising results regarding survival, when used as continuation maintenance (bevacizumab) or as switch maintenance (PARP inhibitors) [5,6]. However, relapses do occur and, the more often they manifest during a patient’s lifetime, the shorter the progression-free survival (PFS) intervals between treatment regimens become [7].

Several subsequent chemotherapeutic regimens, although they are less standardized, are used to treat recurrent ovarian cancer (ROC), considering platinum-based treatment response. As such, patients with platinum-sensitive disease, who relapse after at least 6 months treatment-free interval, can still have a highly effective response to platinum-containing rechallenge (up to 60%), showing a prolonged PFS and overall survival (OS) [8,9,10,11]. Moreover, associating targeted therapy as a maintenance treatment further prolongs PFS in the second-line setting [12]. Regarding the platinum-resistant subgroup, where relapse occurs in the first 6 months, response and survival rates are lower than the opposite when non-platinum therapies are used, a slightly better outcome being observed with the addition of targeted therapies than without them—PFS of 6.7 months for the addition of bevacizumab versus 3.4 months with chemotherapy alone [13]. Multiple chemotherapy drugs are available to be used in cases of recurrences (e.g., pegylated liposomal doxorubicin (PLD), topotecan, gemcitabine, and paclitaxel), with similar efficacy, but with different toxicity profiles [14,15]. However, little information is known about the best selection of therapy in subsequent lines (>second line), to obtain the best long-term result [7,16].

Unlike PFS, which is determined by objective disease progression and death, time-to-treatment-failure (TTF) is an endpoint defined as the time from initiation of treatment to its discontinuation for any reason, including factors unrelated to treatment efficacy (e.g., drug toxicity and patient preference) [17]. The present study aimed to assess the effects of chemotherapy regimens and predictive factors upon TTF from the first line and beyond, in platinum-resistant OC.

## 2. Materials and Methods

### 2.1. Study Design

We developed a single institution-based observational retrospective cohort study on patients diagnosed with OC, treated at the OncoHelp Oncology Center, Timișoara, Romania, from January 2016 to February 2021.

The inclusion criteria for the current study were: patients older than 18 years old, diagnosed with platinum-resistant OC (defined by recurrent disease within 6 months), confirmed by histopathological analysis, who underwent chemotherapy-based treatment with or without anti-angiogenic therapy. Patients with platinum-sensitive disease (defined by recurrent disease after more than 6 months following primary platinum-based treatment) at the beginning of the data collection were excluded.

### 2.2. Data Collection

Clinical assessment and anthropometric and demographic data were collected from medical records, including age, performance status assessment by using Eastern Cooperative Oncology Group score (ECOG PS) and body-mass index (BMI). Hemogram (hemoglobin, neutrophil, and lymphocyte count), biochemical parameters (creatinine), pathological diagnosis, and initial tumor stage were also evaluated.

We measured TTF in days, as the interval from initiation of chemotherapy to its premature discontinuation for any reason, including a decrease in ECOG PS, disease progression, treatment-related adverse events, end of documentation period, patient choice, or death.

### 2.3. Data Analysis

For the assessment of the data collected, we used Epi Info™ (version 7.2.2.6; trademark of the Centers for Disease Control and Prevention, Division of Health Informatics and Surveillance, Center for Surveillance, Epidemiology and Laboratory Services, Atlanta, GA) and MySQL Database Service™ (version 8.0; trademark of the Oracle Corporation). We used descriptive measures, such as percentages and corresponding 95% confidence intervals (CIs). We calculated the measures of central location (means) and dispersion (standard deviation (SD)) for the continuous numeric variables. The statistical analysis for the association of TTF with the demographic, anthropometric, clinical, hematological, biochemical, and therapeutic variables was performed using the Cox Proportional Hazards Survival Regression (CPHSR), test available at https://statpages.info/prophaz.html (Accessed 4 April 2021). We computed the risk ratio (RR) and considered a *p*-value of 0.05 as the threshold for statistical significance.

## 3. Results

We identified 327 female patients treated at the OncoHelp Oncology Center for ovarian cancer, from January 2016 to February 2021, on which we applied the inclusion and exclusion criteria (Figure 1). A remaining total of 78 (23.85%) patients were included in the current study, aged between 43 and 79, with a median age of 62 years old (SD = 8.6).

The background and medical characteristics of the patients included in the study are summarized in Table 1. The most prevalent age group was between 60 and 69 (43.59%), while most of the patients (52, 66.67%) had an ECOG PS equal to 1 at the beginning of the treatment. The distribution according to BMI was similar for normal-weight (25, 32.05%), overweight (26, 33.33%), and obese patients (24, 30.77%).

All patients from the study had a histopathological diagnosis of epithelial ovarian cancer (EOC), serous carcinoma being the most frequent subtype identified (57, 73.08%). Clear cell carcinoma and endometrioid carcinoma were present in low percentages (3.85%, respectively 2.56%), while for 16 patients (20.51%) the subtype of EOC was not specified.

FIGO stage was assessed at the beginning of the treatment, when most of the patients were classified as FIGO stage IIIC (34, 43.59%), followed by FIGO stage IV (23, 29.49%).

After the conversion to platinum-resistant disease, the patients underwent up to nine treatment lines, in decreasing proportions: all of them received the first line of treatment, 51 (65.38%) the second line, 25 (32.05%) the third line, 11 (14.10%) the fourth line, 3 (3.84%) the fifth line, and 1 (1.28%) patient received also from sixth to the ninth line (Figure 2).

Treatment regimens included PLD, bevacizumab, carboplatin, etoposide, gemcitabine, paclitaxel, ifosfamide, and topotecan, alone or in different combinations, as outlined in Table 2 (for lines 1–9). The only patient with nine lines of treatment received carboplatin + paclitaxel in the sixth line, etoposide in the seventh, PLD in the eighth, and ifosfamide in the ninth.

Mean TTF tends to be higher for obese patients (173.45 days) compared to overweight (mean TTF = 149.00 days), normal weight (mean TTF = 146.48 days), and underweight ones (mean TTF = 102.00 days). Regarding the age group, the longest mean TTF (167.34 days) was observed for the patients between 60 and 69 years old, while the shortest mean TTF (121.12 days) was for the group over 70.

Table 3 shows the mean TTF for each treatment regimen and each line of treatment. Adding bevacizumab to the chemotherapy regimen (PLD, carboplatin + paclitaxel, or topotecan) resulted in an increased mean TTF for all the lines in which it was administered (Table 3).

Serous ovarian carcinomas were significantly associated with a prolonged TTF than other histopathological diagnostics, when all treatment lines were taken into consideration (RR = 0.7083, *p* = 0.0479). Analyzing other several factors concerning the TTF in our general cohort, the results revealed that some of them showed a trend towards a positive influence upon it, but without statistical significance: BMI, creatinine level, and neutrophil count (Table 4). On the other hand, a biological factor that may raise concern in TTF was represented by baseline neutrophil values that exceed 6 × 10^9^/L, when taking into consideration all treatment regimens (RR = 2.2148, *p* = 0.0645).

In a subgroup analysis, age and type of chemotherapy seem to contribute to the total duration of treatment. Women older than 60 years of age treated with topotecan presented a prolonged TTF than the opposite age category (RR = 0.5546, *p* = 0.0466), but a different result was seen in women older than 60 years treated with gemcitabine, where age was associated with a higher risk of treatment failure than the younger category, although not statistically significant (RR = 1.5735, *p* = 0.2154). Women under treatment with PLD did not show any significant difference (RR = 1.0882, *p* = 0.77) regarding the age at therapy initiation.

In the subgroup where treatment schedules included bevacizumab, creatinine levels between 0.65 and 1 mg/dL further correlate with improved TTF (RR = 0.1182, *p* = 0.0098). Better outcomes were also observed in PLD-based regimens used in patients who present with the same range for creatinine levels (RR = 0.2930, *p* = 0.0170).

In the cases where treatment regimens with topotecan and PLD included bevacizumab from the first line and beyond, statistically significant results were seen, as expected (RR = 0.3332, *p* = 0.0087 and RR = 0.4187, *p* = 0.0228, respectively). When comparing the chemotherapy sequence between PLD in the first line, followed by topotecan and vice-versa, both in combination with bevacizumab, a trend towards a higher TTF was observed using PLD in the beginning (RR = 0.1277, *p* = 0.0703). Moreover, ages over 60 years old may be considered as a positive factor for TTF when treatment with PLD is used in the first line and beyond. Unfortunately, these results did not prove to be statistically significant.

## 4. Discussion

EOCs have been classified into five major subgroups based on histology, including high-grade serous carcinoma (HGSC; 70%), low-grade serous carcinoma (LGSC; <5%), clear cell carcinoma (CCC; 10–15%), endometrioid carcinoma (EC; 10%), and mucinous carcinoma (MC; 3%), that differ in tumor biology, pathogenesis, molecular alterations, risk factors, and prognosis [18,19,20,21]. In our study, the incidence of the serous carcinoma subtype was the most frequent and significantly associated with a prolonged TTF compared with other histopathological diagnostics (RR = 0.7083, *p* = 0.0479), when all treatment lines were taken into consideration, with the mention that 16 patients (20.51%) had a histopathological diagnosis of EOC, without a subtype specified. This result is concordant with data obtained from previous studies, according to which the OS is higher in the serous subtype compared to the other types [22].

Even though TTF can be influenced by factors unrelated to efficacy (e.g., treatment toxicity and patient preference), it reflects well the clinical applicability of targeted therapies in excessively pretreated patients but is rarely used for regulatory drug approval [17,23,24]. In our study, we considered also the factors unrelated to efficacy when measuring TTF, as the reason for treatment failure or discontinuation was not clearly specified in the medical records. This fact can be considered a limitation of the study, but suggests the need to implement better digital platforms and medical records for collecting data during patients’ treatment, for more complete data to be used during retrospective studies.

Despite important progress that has been made in the treatment of OC, most patients with recurrent ovarian cancer will eventually develop platinum-resistant disease and receive second-line and sometimes several lines of treatment [25]. In general, platinum-resistant patients will be treated with sequential single agents rather than combination therapy, but the benefit of combined therapy and the incorporation of molecular targeted therapies have significantly prolonged the median survival of patients [26]. However, combination chemotherapy may cause significant toxicity with a negative impact on patients’ quality of life [27]. Moreover, the effect of combination chemotherapy on the quality of life of patients with recurrent ovarian cancer has not been sufficiently investigated in clinical trials [28]. It is well known that the benefit of any chemotherapy reduces with each successive line of treatment, as many patients are receiving more than one line of therapy for recurrent disease, usually platinum-based until platinum resistance emerges [25]. In our study, in cases where bevacizumab was added to PLD or topotecan regimens, significantly established bevacizumab as being positively associated with a good total TTF. This fact does not come as a surprise, since it is well known that in the AURELIA trial, consistently improved overall response rate and median PFS when used with topotecan (5.8 months vs. 2.1 months, HR 0.32 95% CI 0.21–0.49) and PLD (5.4 months vs. 3.5 months, HR 0.57, 95% CI 0.93–0.83) [29]. In addition, an interesting fact is that using PLD before topotecan could have been a better choice than vice-versa, including the cases where bevacizumab is added, although statistical significance is missing (RR = 0.1277, *p* = 0.0703). Regarding the influence of age on time to TTF when topotecan-based chemotherapy was used, good results were seen in elderly patients, confirming what Sorio et al. described in the literature [30].

A study performed in 2020 on 283 patients with EOC who were divided into two groups: <65 years (74.6%) group 1 and ≥65 years, group 2 (25.4%), showed that bevacizumab is not associated with an increase in G3/G4 toxicity among the elderly, suggesting that age is not a predictive factor of adverse events for those receiving bevacizumab [31]. In our study, all treatment administrations that included bevacizumab, creatinine levels between 0.65 and 1 mg/dL further correlate with improved TTF (RR = 0.1182, *p* = 0.0098).

The influence of body mass index on the risk and prognosis of ovarian cancer is subject to a lot of debate, as underweight or obese patients are considered to be a group at risk. In this group finding the right treatment seems to be more challenging, as chemotherapy-associated toxicity is one of the limiting factors regarding treatment response, patient outcome, and quality of life [32,33,34]. To support this, a study published in 2018 aimed to evaluate the impact of BMI on toxicity in patients undergoing chemotherapy. Their results showed BMI-associated differences in the number of administered cycles. The most important aspect was related to the fact that patients with lower BMI received fewer chemotherapy cycles in comparison to those with higher BMI [35]. Another study published in 2014 found that a lower BMI in patients with advanced-stage ovarian cancer is considered to be an early marker for poor prognosis and treatment discontinuation, as well accompanied by decreased immunity and advanced tumor inflammation [36]. A study published in 2015 suggested that body size should not be a major factor influencing dose reduction decisions in women with ovarian cancer [37]. Despite controversies regarding the influence of BMI on the risk and prognosis of high-grade serous ovarian cancer, there is very limited data available on BMI-related toxicities regarding systemic therapy [38,39]. In our study, although in the descriptive analysis we observed increased mean values of TTF being associated with increased BMI, statistical significance did not show proof in this situation.

Nevertheless, our study has its limitations: the number of patients included was relatively small, with disproportions due to the nature of a typical non-randomized study. Also, complete blood count and renal function test results were not available before December 2018, so analyses that included these assessments used fewer patients.

In conclusion, a few positive predictive factors were identified in our study: serous carcinoma subtype, age over 60 years old for patients treated with topotecan with and without bevacizumab, creatinine levels between 0.65 and 1 mg/dL (for patients treated with PLD and regimens that contain bevacizumab), and the inclusion of bevacizumab in combination with PLD or topotecan as the physician’s choice of treatment from the first line and beyond. Further research based on larger populations would need to be performed to assess other factors’ statistical significance in the influence upon TTF.

## Figures and Tables

**Figure 1 jcm-10-03596-f001:**
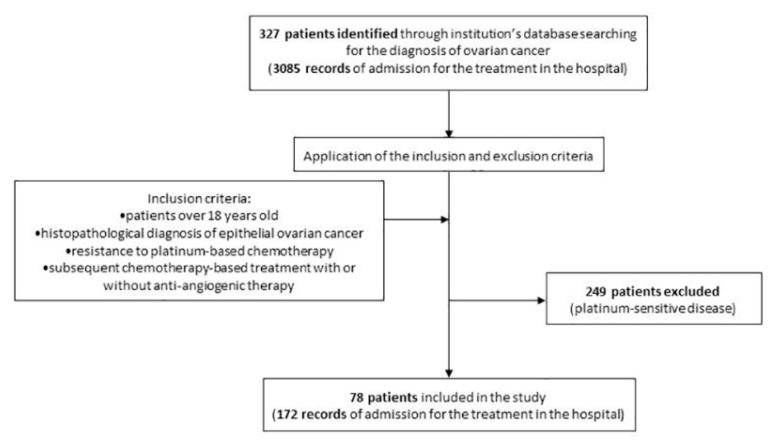
Study design flowchart and application of inclusion/exclusion criteria.

**Figure 2 jcm-10-03596-f002:**
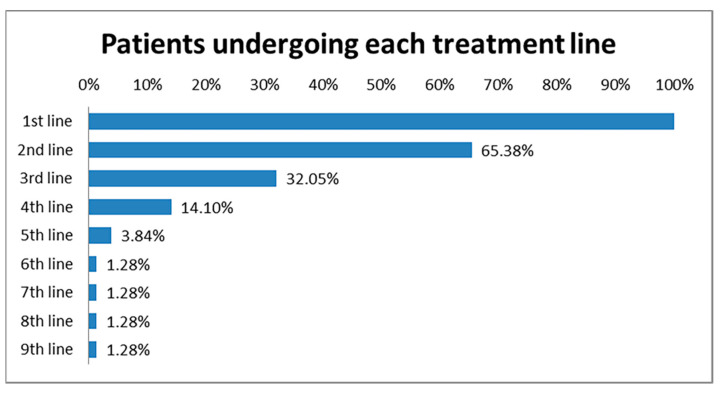
Percentages of patients undergoing each treatment line, after the conversion to platinum-resistant disease.

**Table 1 jcm-10-03596-t001:** Background and medical characteristics of the patients undergoing the first line of treatment: age, ECOG PS, BMI, the subtype of EOC, and initial disease FIGO stage.

Background Characteristic	*n*	% (Out of N)
**Age (N = 78)**		
40–49	9	11.54%
50–59	22	28.21%
60–69	34	43.59%
Over 70	13	16.67%
**ECOG PS (N = 78)**		
0	19	24.36%
1	52	66.67%
2	7	8.97%
**BMI (N = 78)**		
Below 18.5 (Underweight)	3	3.85%
18.5–24.9 (Normal weight)	25	32.05%
25.0–29.9 (Overweight)	26	33.33%
Above 30.0 (Obesity)	24	30.77%
**Subtype of EOC (N = 62)**		
Clear cell carcinoma	3	3.85%
Endometrioid carcinoma	2	2.56%
Serous carcinoma	57	73.08%
Not specified	16	20.51%
**Initial disease stage (N = 78)**		
FIGO stage IA	1	1.28%
FIGO stage IC	3	3.85%
FIGO stage IIA	2	2.56%
FIGO stage IIB	1	1.28%
FIGO stage IIIA	8	10.26%
FIGO stage IIIB	6	7.96%
FIGO stage IIIC	34	43.59%
FIGO stage IV	23	29.49%

EGOG PS = Eastern Cooperative Oncology Group score for Performance Status; BMI = body-mass index; EOC = epithelial ovarian cancer; FIGO = Féderation Internationale de Gynécologie et d’Obstétrique.

**Table 2 jcm-10-03596-t002:** Number and percentage of patients for each treatment regimen (PLD, Bevacizumab, Carboplatin, Etoposide, Gemcitabine, Paclitaxel, Ifosfamide, and Topotecan, alone or in different combinations) in lines 1 to 9.

Treatment Regimen	Line of Treatment							
	First Line	Second Line	Third Line	Fourth Line	Fifth Line	Seventh Line	Eighth Line	Ninth Line
	(N = 78)	(N = 51)	(N = 25)	(N = 11)	(N = 3)	(N = 1)	(N = 1)	(N = 1)
PLD	30.77% (24)	19.61% (10)	8.00% (2)	0.00% (0)	0.00% (0)	0.00% (0)	0.00% (0)	0.00% (0)
PLD + Bevacizumab	14.10% (11)	5.88% (3)	0.00% (0)	0.00% (0)	0.00% (0)	0.00% (0)	0.00% (0)	0.00% (0)
PLD + Carboplatin	1.28% (1)	0.00% (0)	0.00% (0)	0.00% (0)	0.00% (0)	0.00% (0)	0.00% (0)	0.00% (0)
Carboplatin	1.28% (1)	3.92% (2)	0.00% (0)	0.00% (0)	0.00% (0)	0.00% (0)	0.00% (0)	0.00% (0)
Carboplatin + Etoposide	0.00% (0)	1.96% (1)	4.00% (1)	0.00% (0)	0.00% (0)	0.00% (0)	0.00% (0)	0.00% (0)
Carboplatin + Gemcitabine	0.00% (0)	0.00% (0)	8.00% (2)	0.00% (0)	0.00% (0)	0.00% (0)	0.00% (0)	0.00% (0)
Carboplatin + Paclitaxel	0.00% (0)	5.88% (3)	16.00% (4)	18.18% (2)	33.33% (1)	100.00% (1)	0.00% (0)	0.00% (0)
Carboplatin + Paclitaxel + Bevacizumab	2.56% (2)	1.96% (1)	0.00% (0)	0.00% (0)	0.00% (0)	0.00% (0)	0.00% (0)	0.00% (0)
Etoposide	0.00% (0)	0.00% (0)	0.00% (0)	18.18% (2)	0.00% (0)	0.00% (0)	100.00% (1)	0.00% (0)
Gemcitabine	11.54% (9)	25.49% (13)	28.00% (7)	54.55% (6)	0.00% (0)	0.00% (0)	0.00% (0)	0.00% (0)
Ifosfamide	0.00% (0)	0.00% (0)	0.00% (0)	0.00% (0)	0.00% (0)	0.00% (0)	0.00% (0)	100.00 (1)
Ifosfamide + Paclitaxel	0.00% (0)	3.92% (2)	0.00% (0)	0.00% (0)	33.33% (1)	0.00% (0)	0.00% (0)	0.00% (0)
Paclitaxel	0.00% (0)	0.00% (0)	0.00% (0)	0.00% (0)	33.33% (1)	0.00% (0)	0.00% (0)	0.00% (0)
Topotecan	30.77% (24)	23.53% (12)	32.00% (8)	9.09% (1)	0.00% (0)	0.00% (0)	0.00% (0)	0.00% (0)
Topotecan + Bevacizumab	7.69% (6)	7.84% (4)	4.00% (1)	0.00% (0)	0.00% (0)	0.00% (0)	0.00% (0)	0.00% (0)

PLD = pegylated liposomal doxorubicin.

**Table 3 jcm-10-03596-t003:** Mean TTF (days) for each treatment regimen and each line of treatment.

Treatment Regimen	Line of Treatment								
	First Line	Second Line	Third Line	Fourth Line	Fifth Line	Sixth Line	Seventh Line	Eighth Line	Ninth Line
PLD	140.83	101.30	117.00	-	-	-	-	91.00	-
PLD + Bevacizumab	195.54	393.66	-	-	-	-	-	-	-
PLD + Carboplatin	474.00	-	-	-	-	-	-	-	-
Carboplatin	148.00	254.50	-	-	-	-	-	-	-
Carboplatin + Etoposide	-	21.00	98.00	-	-	-	-	-	-
Carboplatin + Gemcitabine	-	-	204.50	-	-	-	-	-	-
Carboplatin + Paclitaxel	-	206.33	139.75	222.50	359.00	116.00	-	-	-
Carboplatin + Paclitaxel + Bevacizumab	202.50	303.00	-	-	-	-	-	-	-
Etoposide	-	-	-	144.50	-	-	91.00	-	-
Gemcitabine	124.33	111.00	105.85	136.83	-	-		-	-
Ifosfamide	-	-	-	-	-	-	-	-	143.00
Ifosfamide + Paclitaxel	-	103.50	-	-	260.00	-	-	-	-
Paclitaxel	-	-	-	-	173.00	-	-	-	-
Topotecan	121.29	117.08	133.62	52.00	-	-	-	-	-
Topotecan + Bevacizumab	236.16	409.50	150.00	-	-	-	-	-	-

TTF = time-to-treatment- failure; PLD = pegylated liposomal doxorubicin.

**Table 4 jcm-10-03596-t004:** Association of TTF with the demographic (age), anthropometric (BMI), clinical (ECOG PS), hematological (Hb, ANC, ALC), biochemical (Cr), and therapeutic variables (adding of anti-angiogenic therapy-Bevacizumab), performed using the Cox Proportional Hazards Survival Regression.

N = 56 *	RR (95% CI)	*p*-Value
Age (years) (average 62.0178)	1.0363 (0.9926–1.0819)	0.1047
BMI (kg/m^2^) (average 28.5132)	0.9349 (0.8673–1.0079)	0.0792
ECOG PS (from 0 to 2)	2.0042 (0.5301–7.5765)	0.3055
Hb (g/dL) (average 11.225)	0.7913 (0.6252–1.0015)	0.0515
ANC (×10^9^/L) (average 4.4875)	1.0847 (0.9364–1.2565)	0.2785
ALC (10^9^/L) (average 1.5771)	0.8908 (0.4425–1.7934)	0.7461
Cr (mg/dL) (average 0.9023)	0.2451 (0.0582–1.0327)	0.0553
ChT + Bevacizumab (17/56)	0.5782 (0.2703–1.2365)	0.1578

BMI = body-mass index; EGOG PS = Eastern Cooperative Oncology Group score for Performance Status; Hb = hemoglobin level; ANC = absolute neutrophil count; ALC = absolute lymphocyte count; Cr = creatinine level; ChT = chemotherapy. * The number of patients with the complete data mentioned in the table.

## Data Availability

The data presented in this study are available on request from the corresponding author. The data are not publicly available due to hospital policy.

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
