# Peer review of "Treatment Experience and Predictive Factors Associated with Response in Platinum-Resistant Recurrent Ovarian Cancer: A Retrospective Single-Institution Study"

_jcm, 2021, doi:10.3390/jcm10163596_

Round 1

Reviewer 1 Report

The author's have addressed my prior critiques adequately.

Author Response

Dear reviewer,

Thank you for your consideration. We made corrections regarding English language and style.

Kind regards!

Reviewer 2 Report

Authors assessed the clinical and biological factors for time-to-treatment failure (TTF) in patients being treated with various next-line chemotherapeutic regimens in the treatment of platinum-resistant recurrent ovarian cancer. They were able to identify statistically significant positive predictive factors for TTF based on histologic subtype, age with certain treatments, creatinine values with certain treatments, and for certain specific treatments.

Limitations of the study were provided. However, there were several factors that did not reach statistical significance that the authors appeared to draw conclusions from. Additionally, the definition of TTF and factors used to assess this were somewhat misleading.

Please see the comments below.

Concern 1: The definition used in the manuscript for TTF is the time from initiation of treatment to its discontinuation for any reason, including factors unrelated to treatment efficacy, and it was measured as the interval from initiation of chemo to its premature discontinuation for any reason (including efficacy – eg disease progression). However in the discussion, it states that TTF is an endpoint influenced by factors unrelated to efficacy. Please clarify that TTF includes both factors unrelated and related to efficacy.

Concern 2: Authors stated that several other factors could positively influence the outcome (eg BMI, creat level, neutrophil count, using PLD before topotecan and age), but these were not statistically significant. As significance could not be detected with this cohort of patients, conclusions like this should not be drawn. It can be modified to state that there was a trend towards positive influence, but a larger study would need to be performed to assess the statistical significance.

Author Response

Dear reviewer,

We appreciate your observations and we tried to clarify the issue regarding the TTF in the "Discussion" section. Also, we rephrased the sentences that refer to statistically insignificant factors, in the "Results" sector. In the "Conclusion" sector, we only referred to results that were statistically significant, even though most of them were analyzed on subgroups of patients.

Kind regards!

This manuscript is a resubmission of an earlier submission. The following is a list of the peer review reports and author responses from that submission.

Round 1

Reviewer 1 Report

Major concerns:

The manuscript  requires extensive editing of english language and style.

In addition, at the mansucript is confusingly written. At several points different lines of toughts seem to be mixed.

minor concerns:

Table 1: Whats the point in giving 95%-CI within a table depicting descriptive statstics?

Reviewer 2 Report

My review for the manuscript JCM-1243058 are as below. 

The manuscript (jcm-1243058) by Radu Dragomir et al, had investigated the

 treatment experience and predictive factors associated with response in platinum-resistant recurrent ovarian cancer.

My specific comments are as follows-

  1. The introduction could be improved and the objective of the study was not properly described.  The authors had identified positive predictive factors for TTF related to anthropometry , renal function and treatment choice.
  2. The material method portion needs significant improvement.
  3. The observation and result presentation are extensive.

The manuscript should be minor revision, mainly formatting.

Reviewer 3 Report

The single-institution retrospective analysis provided by Dragomir et al. aims to identify the effects of chemotherapy regiments and predictive factors in platinum-resistant recurrent OC using time-to-treatment-failure (TTF) as the primary outcome measure. Factors associated with prolonged TTF include the serous subtype of epithelial ovarian carcinoma in all treatment lines (RR = 0.7083, p = 0.0479), >60 years old treated with topotecan (RR = 0.5546, p = 0.0466), creatinine levels between 0.65-1 mg/dl treated with bevacizumab (RR = 0.1182, p = 0.0098) and PLD-based regimens (RR = 0.2930, p = 0.0170), and topotecan (RR = 0.332, p = 0.0087) and PLD-based (RR = 0.4187, p = 0.0228) treatment regimens that included bevacizumab. Other factors and treatment lines investigated did not show statistical significance, though the authors discuss some potential “trends,” included >60 years old treated with gemcitabine (RR = 1.5735, p = 0.2154), BMI (RR = 0.0792, p = 0.0792), creatinine level (RR = 0.2451, p = 0.0553), neutrophil count (RR = 0.7913, p = 0.0515), and neutrophil values that exceed 6 x 10^9/l in all treatment levels (RR = 2.2148, p = 0.0645).

The enthusiasm for this study is dampened by the fact that few factors investigated were statistically significant (potentially due to small sample size, which the authors acknowledge in lines 276-279) and many of its results have been already previously reported. The use of bevacizumab in PLD-based and topotecan treatments regimens was demonstrated in the AURELIA trial, which the authors acknowledge (lines 239-245). The authors also acknowledge that the effect of age when using topotecan-based chemotherapy has been previously reported (lines 248-250). The authors’ discussion that these results have been previously published is appreciated, and validation of published results are still worth reporting. Here are suggestions to improve the strength of the study for publication:

  1. The wording and structure of sentences are confusing at times (e.g., the sentence that spans lines 74-79 and the discussion on bevacizumab treatment from lines 203-209). It may benefit to have an English-writing proofreader.

  2. Statistically significant results are sometimes buried in paragraphs with potential “trends.” It would be clearer to first discuss all statistically significant results and then discuss all “trends.”

  3. Figure 3 is difficult to understand in its current form. For clarity, the y-axis needs to range from 0-60% (not 0-100%) so the bars can be visualized more clearly. Furthermore, it would be advantageous for the x-axis to consist of 1st line, 2nd line, 3rd line, etc. and graph the different regimens as bars. This also needs to include lines 6-9, even though it is only one patient (if that patient is included in the analysis, they should be included in the graph). It does appear that table 2 serves this purpose, in which case perhaps this graph (figure 3) can be omitted outright.

  4. Table 2 needs to include the one patient with treatment lines 6-9 so that the whole patient population analyzed can be properly visualized.

  5. In table 4, the sample size (n) for each factor (age, BMI, etc.) should be reported. In the discussion (lines 278-279), it is mentioned that complete blood count (CBC) and renal function test results were only available after December 2018, suggesting that the n for each of the factors listed in table 4 is different from the total number of patients analyzed in the study.

  6. The association between the serous histological subtype and prolonged TTF is a bit surprising, given the lethality of platinum-resistant high-grade serous ovarian carcinoma (HGSC). It would help to have more discussion as to why serous ovarian cancer might have prolonged TTF. Also, what happens if this result is separated between high-grade and low-grade serous ovarian carcinoma?

  7. As the authors point out (lines 225-227), TTF is not a commonly used endpoint and may be influenced by factors unrelated to treatment regimen. TTF needs to be introduced in the introduction with discussion on why TTF is the most appropriate outcome measure to use in this study and how it differs from the more commonly used outcome measure progression-free survival (PFS). Furthermore, the conclusion needs a discussion on what unrelated factors can influence TTF and why it is still the most appropriate outcome measure to use in this study, regardless of these limitations.

  8. The authors spend a significant amount of space discussing “trends” (p > 0.05). It would be prudent to discuss that all “trends” produced from the analysis need to be further investigated, likely with larger sample sizes, to prove statistical significance.

  9. The abstract and conclusion (lines 280-284) are missing serous ovarian carcinoma as a positive predictive factor.